# Capturing Information About Multiple Sclerosis Comorbidity Using Clinical Interviews and Administrative Records: Do the Data Sources Agree?

**DOI:** 10.3390/healthcare13111281

**Published:** 2025-05-28

**Authors:** Michela Ponzio, Maria Cristina Monti, Paola Borrelli, Giulia Mallucci, Daniela Amicizia, Filippo Ansaldi, Giampaolo Brichetto, Marco Salivetto, Andrea Tacchino, Pietro Perotti, Simona Dalle Carbonare, Roberto Bergamaschi, Cristina Montomoli

**Affiliations:** 1Scientific Research Area, Italian Multiple Sclerosis Foundation, 16149 Genoa, Italy; giampaolo.brichetto@aism.it (G.B.); marco.salivetto@aism.it (M.S.); andrea.tacchino@aism.it (A.T.); 2Unit of Biostatistics and Clinical Epidemiology, Department of Public Health, Experimental and Forensic Medicine, University of Pavia, 27100 Pavia, Italy; cristina.monti@unipv.it (M.C.M.); paola.borrelli@unich.it (P.B.); cristina.montomoli@unipv.it (C.M.); 3Laboratory of Biostatistics, Department of Medical, Oral and Biotechnological Sciences, University “G. d’Annunzio” Chieti-Pescara, 66100 Chieti, Italy; 4Multiple Sclerosis Research Centre, IRCCS Mondino Foundation, 27100 Pavia, Italy; giulia.mallucci@gmail.com (G.M.); roberto.bergamaschi@mondino.it (R.B.); 5Department of Health Sciences, University of Genoa, 16132 Genoa, Italy; daniela.amicizia@alisa.liguria.it (D.A.); filippo.ansaldi@alisa.liguria.it (F.A.); 6A.Li.Sa., Liguria Health Authority, 16121 Genoa, Italy; 7AISM Rehabilitation Centre Liguria, Italian Multiple Sclerosis Society, 16149 Genoa, Italy; 8Pavia Health Protection Agency, 27100 Pavia, Italy; pietro_perotti@ats-pavia.it (P.P.); simona_dalle_carbonare@ats-pavia.it (S.D.C.)

**Keywords:** comorbidity, multiple sclerosis, administrative healthcare data, clinical interview, validation

## Abstract

**Background/Objectives:** Multiple sclerosis (MS) is often associated with comorbidities that affect clinical outcomes. Data on comorbidities can be sourced from self-reports, medical records, and administrative databases. The gold standard for collecting such data is prospective clinical collection, as in clinical trials, but this is not feasible in large epidemiological studies. This study aimed to assess the agreement between two data sources, clinical interviews and administrative records, identifying major comorbidities in people with MS (pwMS). **Methods**: We evaluated the agreement between clinical interview data and administrative records in pwMS enrolled at two sites (2021–2022). Seven comorbidities were investigated: depression, anxiety, diabetes, hypertension, autoimmune disease, chronic lung disease, and hyperlipidemia. We used kappa (κ), sensitivity, specificity, and predictive values to assess agreement. **Results**: The frequency of comorbidities varied between the sources. Administrative data often underestimated hypertension, autoimmune diseases, hyperlipidemia, and anxiety, but over-reported depression. It had high sensitivity for diabetes (80%) and moderate sensitivity for hypertension (62%). The agreement for diabetes (κ = 98.9%, PABAK = 0.98, positive agreement = 83.3%) and hypertension (κ = 89.8%, PABAK = 0.80, positive agreement = 70.8%) was high. **Conclusions**: The agreement between administrative data and clinical interviews was excellent for diabetes and hypertension. For other conditions, such as psychiatric, hyperlipidemia, and autoimmune comorbidities, administrative data had lower sensitivity, and often under-reported or misclassified the data.

## 1. Introduction

Multiple sclerosis (MS) is a chronic, inflammatory, and neurodegenerative disorder of the central nervous system. It is estimated that 127,000 people in Italy and 2.8 million worldwide live with MS [1]. Increasing evidence suggests that comorbidities are common in people with MS (pwMS), and can adversely affect their clinical outcomes [2], negatively impacting the disease course, delaying diagnosis, allowing progression of disability, and negatively affecting treatment management and adherence [3,4]. Given the high prevalence of comorbid conditions in MS and the potential for their prevention or treatment, comorbidities are attracting growing interest as a factor that might explain the heterogeneity of outcomes in this condition [5]. Physical and mental comorbidities not only influence outcomes but also strongly affect quality of life in pwMS [6].

Complete knowledge of comorbidities is essential in order to be able to detect health needs and correctly plan coordinated strategies across multiple levels and healthcare sectors [7]. Two recent studies conducted in Italy have highlighted that underestimating comorbidities has significant health, social, and economic consequences [8], profoundly affecting the quality of life and work-related activities of pwMS [9]. It also allows optimal risk stratification of individual patients and therefore personalized therapies [10,11].

Potential sources of comorbidity data include self-reports, medical records, and administrative databases. It was observed that the validity of self-reported comorbidity data in pwMS varies depending on the specific condition [12]. Historically, clinical interviews have been the reference method for gathering information about pre-existing comorbidities; however, being time consuming and costly, they are not feasible in large epidemiological studies. Administrative databases, on the other hand, are an accessible, cost-effective source of data that, covering large populations, can potentially help to answer research questions that cannot be addressed through interviews or reviews of records; however, they are sometimes incomplete and not completely reliable. Previous research has shown that coded comorbidities in administrative databases show varying levels of agreement with medical record review data. The authors reported that administrative databases tend to underestimate the prevalence of some comorbid conditions; asymptomatic conditions were noted to have lower levels of agreement, but the overall agreement with the patient chart review was very good [13].

While comorbidities are a significant concern in MS, there are only a few validated approaches for their evaluation. Only Marrie and collaborators have tried to validate administrative definitions for several comorbidities of potential importance in MS and to describe their prevalence among persons with MS [14].

The aim of this study was to quantify the agreement between data on major comorbidities derived from two different data sources (clinical interviews and administrative records) in a sample of pwMS.

## 2. Materials and Methods

A sample of pwMS was recruited at the MS Center of the Mondino Foundation, Pavia, and at the AISM (Italian Multiple Sclerosis Association) rehabilitation service in Liguria, Genoa, between 2020 and 2021. The eligibility criteria included a confirmed MS diagnosis [15], an age of 18 years or older, and official registration as a patient within the administrative areas of Pavia or Genoa. Prior to participation, all individuals provided written informed consent in compliance with the revised Declaration of Helsinki. Clinical interviews were conducted by research assistants trained in collecting data about the presence, in the previous 10 years, of the seven main comorbidities in MS [4]: depression, anxiety, diabetes, hypertension, autoimmune disease, chronic lung disease, and hyperlipidemia. In order to improve the quality of the data collected, the research assistants, before conducting the interviews, retrospectively reviewed the medical records of the pwMS included in the study using a structured clinical research form (CRF) (Appendix A) including physicians’ notes and prescribed medications.

Each condition was classified as present or absent based on any documented statement by a physician in the medical record, following the specific definitions provided in the CRF. Then, the research assistants used the data collected in the CRF to orient and stimulate the clinical interviews, and also to reduce the recall bias of the patients.

Where there was a conflict between the clinical interview and retrospective review of the medical record, priority was given to the medical record. The CRF also captured demographics (age, sex, educational level, marital status, living situation, and city of residency) and other clinical information (diagnosis and onset date, disease course, disease duration, and disability levels quantified according to Kurtzke’s Expanded Disability Status Scale (EDSS) [16]). A short questionnaire is presented as Appendix A to show that the variables collected in the CRF are useful for sample characterization and to quantify the comorbidity presence. During the same period, the two participating local health authorities, namely the Liguria Health Authority (Azienda Ligure Sanitaria, A.Li.Sa) and the Pavia Health Protection Agency (Agenzia di Tutela della Salute, ATS), searched their administrative databases for the presence of coded comorbidities in the pwMS enrolled in the study; the pwMS themselves were identified through record linkage, where each individual’s tax code (anonymized) was used as a unique personal identification code. The presence of the aforementioned comorbid conditions was identified through a surveillance method that local health authorities use to monitor the prevalence of chronic diseases. Essentially, current databases are integrated into a single population database (Banca Dati Assistiti, BDA); disease-specific algorithms implemented in the BDA then combine data on inpatient diagnoses/procedures [as defined using the International Classification of Diseases (ICD-9-CM)], disease-specific payment exemptions [ESE codes], and outpatient delivery of drugs [coded according to the Anatomical Therapeutic Chemical (ATC) Classification System] and health services [SER codes], all systematically collected in the available regional administrative databases.

For just two comorbidities, anxiety and depression, which are not routinely monitored by the BDA system, we used an algorithm developed ad hoc on the basis of use of specific drugs. The Appendix A reports the disease-specific and ad hoc algorithms used to identify the presence of the comorbid conditions studied.

### Statistical Analysis

Comorbidity frequencies and 95% confidence intervals (95% CIs) were calculated using both data sources. McNemar’s test for paired proportions was used to assess comorbidity frequency differences by data source. To assess agreement, on the presence of comorbidities, between the clinical and the administrative data sources, we considered Cohen’s kappa coefficient (κ), a bias index (BI), a prevalence index (PI), a prevalence-adjusted bias-adjusted kappa (PABAK), along with positive and negative percentage agreement. The kappa coefficient is a standardized value representing chance-adjusted agreement, with 1 indicating perfect agreement and 0 representing no agreement [17]. A common criticism of κ is its high dependence on the prevalence of the condition in the population [18,19]. To address this limitation, Byrt and coworkers [20] proposed the PABAK, which assumes a fifty percent prevalence of the condition and the absence of any bias. Compared with κ, PABAK better reflects the ideal situation, disregarding the variation in prevalence across the conditions and the bias present in the “real” world.

A κ greater than 0.80 was considered to indicate excellent agreement, values between 0.61 and 0.80 suggested substantial agreement, values in the range 0.41–0.60 indicated moderate agreement, values in the range 0.20–0.40 signified fair agreement, and values less than 0.20 reflected poor agreement [21]. To calculate the 95% CIs for κ and PABAK, 200 bootstrap replications were performed.

To enhance the interpretation of κ, positive and negative percentage agreements (proportions of specific agreement) were calculated for each comorbidity. These statistics account for the potential inflation of agreement when the condition under study is rare. A positive percentage agreement estimates the proportion of positive agreement (i.e., presence of a comorbidity in both sources) relative to the average number of positive readings (i.e., average number of comorbidities from both sources), while a negative percentage agreement estimates the proportion of negative agreement (i.e., no comorbidities in either source) relative to the average number of negative readings (i.e., average number of patients with no comorbidities in both data sources) [18]. Finally, sensitivity, specificity, and positive/negative predictive values were calculated by designating the clinical interviews as the gold standard. A stratified analysis was performed for each condition based on sex, age (≤50 and >50 years), and disability level, as measured by the EDSS score (<3.5 indicating mild disability, and ≥3.5 indicating moderate to severe disability). Differences across strata were assessed using tests of proportions, and sensitivity and positive predictive values were indicators of specific focus. All analyses were conducted using Stata Version 17 (Stata-Corp, College Station, TX, USA).

## 3. Results

The analysis was conducted in the 635 pwMS for whom comorbidity data were present in both the data sources considered. Women (*n* = 409) accounted for 64.4%, and the mean age of the sample was 49.7 ± 12.1 years (age range: 19–84). The most common disease phenotype was relapsing-remitting (70.6%), followed by secondary progressive (22.5%) and primary progressive (6.9%); the mean Expanded Disability Status Scale score was 3.6 ± 2.2, and the mean disease duration was 17.2 ± 10.7 years.

Based on the administrative data collected by the two local health authorities, 53.4% of the sample had at least one comorbidity, versus the 47.4% with at least one comorbidity identified through a clinical interview conducted at either of the two participating MS centers (*p* < 0.001). Depression was found to be more frequent in the administrative records than in the clinical interview-derived data, whereas hypertension, autoimmune disease, hyperlipidemia, and anxiety were all under-reported in the administrative records with respect to the clinical data source. Substantially significant differences were observed when comparing the frequencies of the following comorbidities between the two sources: depression (39.7% vs. 16.1%, *p* < 0.001), anxiety (2.7% vs. 12.3%, *p* < 0.001), and hypertension (15.1% vs. 20.3%, *p* < 0.001); in the case of both autoimmune disease (5.2% vs. 9.3%, *p* < 0.001) and hyperlipidemia (3.2% vs. 6.5%, *p* = 0.007), the difference was less evident but still significant (Figure 1).

While a comparison of frequency data reveals the extent to which the two data sources detected the different conditions, it does not show whether they identified the same patients. To determine the extent to which the clinical and administrative sources identified the same patients, the agreement between them was evaluated for each condition. As reported in Figure 2, the distribution patterns varied greatly between the different comorbidities. Both sources captured diabetes and hypertension in a high proportion of patients. With regard to the other comorbidities, the administrative data captured a higher proportion of patients with depression, whereas the presence of anxiety, hyperlipidemia, autoimmune disease, and to a lesser extent, chronic lung disease, was more often identified from the clinical data.

The bars show the percentages of subjects captured by two different sources relative to the total number of patients for each comorbidity, and therefore the different distribution patterns. Black shows the level of agreement between the sources, light gray the proportion of patients identified only by the clinical data, and dark gray the proportion identified only by the administrative data.

Table 1 shows the measures of agreement calculated between the two data sources that were analyzed. κ ranged from 0.02 (poor agreement) for anxiety to 0.85 (excellent agreement) for diabetes, and PABAK from 0.33 (depression) to 0.98 (diabetes). The PABAK values were higher than the κ ones for all the conditions evaluated. Overall, we observed a minor bias effect (the BI ranged from 0 to 0.24), but a considerable prevalence effect (the PI ranged from −0.95 to −0.44). The κ and PABAK values varied significantly for several specific comorbidities, suggesting that this statistic is sensitive to change due to sparse data on these conditions. For hyperlipidemia and anxiety, poor agreement (κ < 0.20) was found between the clinical records and the administrative data, but after adjusting for prevalence and bias, the agreement seems to improve. Likewise, chronic lung disease and autoimmune disease showed agreement ranging from fair (κ: 0.31 and 0.39, respectively) to excellent (PABAK ≥ 0.84). Both κ and PABAK values demonstrated excellent or substantial agreement between the data sources for diabetes and hypertension, and fair agreement for depression. Since the prevalence effect was very large, we calculated positive and negative agreement to enhance the interpretation of κ. Most comorbidities exhibited very good negative agreement. Diabetes and hypertension were the only two comorbidities that showed, respectively, very good (83.3%) and good (70.8%) positive agreement. The other positive agreement values ranged from 43.5% for autoimmune disease to 6.3% for anxiety.

Finally, on the basis of the assumption that the clinical data correctly identified the presence/absence of the conditions, sensitivity, specificity, and predictive values were calculated (Table 2). The results showed that diabetes, depression, and hypertension were more likely to be captured by the administrative data source when the clinical interview data also showed them to be present (with good sensitivity). It is worth noting the rather low positive predictive value for depression (28.2%), which seems to suggest that little evidence of this mood disorder emerged during clinical interviews, even though this comorbidity was present in the administrative database.

Associations between participant characteristics and the strength of agreement between administrative databases and clinical interviews are reported in the Appendix A. Regarding the various comorbidities, we found that male sex was significantly associated with higher agreement for both hypertension and chronic lung disease, while older age was associated with higher agreement only for hypertension. A mild level of disability was linked to greater agreement for hyperlipidemia compared to moderate/severe levels; however, the sensitivity and positive predictive value were not clinically meaningful.

## 4. Discussion

This study analyzed the level of agreement between data of major comorbidities (depression, anxiety, diabetes, hypertension, autoimmune disease, chronic lung disease, and hyperlipidemia) in people with MS obtained from administrative databases and from clinical interviews. The aim was to establish whether administrative databases, as an alternative to clinical interviews, can provide reliable and cost-effective data that, moreover, cover large populations. It is important to underline that the clinical interview used in our study to collect information about comorbidities in pwMS was a tool halfway between self-reported data and medical record reviews. Indeed, to drive the interviews, the research assistants retrospectively reviewed the medical records of the pwMS. A prospective data collection with a priori endpoints, as occurs in clinical trial settings, would be the ideal gold standard for comparison.

Before interpreting our results, it is necessary to underline some important differences between the two data acquisition methods. First of all, they differ in terms of their objectives. Clinical interviews, focusing on patients’ clinical histories and specific comorbidities, are study-specific. Administrative databases, on the other hand, have a much broader purpose, and the standard coding systems used (ICD-9-CM or the ATC classification) do not always reflect the real-world clinical condition. In addition, there may be differences in the detection of clinically identified versus administrative database-coded comorbidities. In this study, the clinical interviews primarily addressed the history of coexisting conditions, whereas the administrative data appeared to be more sensitive in detecting active conditions. While no inter-rater reliability evaluation was undertaken to determine the consistency of medical record reviews, data were collected using a standardized data collection form, and the reviewer was medically trained. It is also important to note that coding errors in the administrative databases may have contributed to the under-ascertainment of certain comorbidities. The adoption of updated classification systems (e.g., ICD-10-CM) could enhance data quality and specificity, particularly for conditions such as autoimmune or mental health disorders. Another critical point concerns the relatively small size of the present study sample, which led to low rates of some comorbidities, such as chronic lung disease. Finally, this study was conducted in only two centers, and therefore, the findings may not be generalizable to all other Italian settings. However, the selected centers—a third-level MS center in Pavia and the AISM Rehabilitation Service of Liguria in Genoa—are key local reference institutions, likely offering a representative sample of the broader MS population.

Our results showed that the frequency of single comorbidities varies depending on the data source. Overall, the administrative data tended to underestimate the occurrence of some comorbid conditions (hypertension, autoimmune diseases, hyperlipidemia, and anxiety), but over-reported depression. Under-reporting by administrative databases is consistent with the results obtained in other studies, and also in different populations, and may be due to a tendency to include more symptomatic than asymptomatic conditions [22,23]. However, other studies revealed fair agreement for depression. In particular, the results for depression could be a reflection of the under-diagnosis of mental health issues linked to individuals’ unwillingness to seek treatment for them [24,25].

It is worth considering that the anxiety and depression results may be an indication that the algorithm used (based only on the use of specific drugs) struggles to identify these comorbidities correctly from administrative data. In particular, the over-reporting of depression in the administrative dataset could be explained by the use of antidepressant drugs for other conditions (e.g., anxiety or sleep disorders). Although the case definition for anxiety and depression still needs to be validated in the MS population, a potential improvement to the algorithm to increase the identification precision could be accomplished by incorporating diagnostic codes, repeated prescription criteria, or psychiatric visit data. The variations in frequency observed between the data sources were reflected in the κ and PABAK statistics. As expected, the difference between these two statistical values for single conditions decreased as their prevalence increased [20]. While the bias effects were relatively minor overall, the prevalence effect was substantial. Therefore, following Byrt’s recommendation [20], we reported both positive and negative agreement to better interpret κ. We observed good agreement between the clinical and administrative data sources where neither of these showed a comorbidity. By contrast, when comorbidities were found in both, the level of agreement was generally relatively poor; exceptions were diabetes and hypertension, for which we found good positive agreement (κ) and substantial agreement (PABAK).

Consistent with the findings of others [26], our study showed that the administrative data had a good capacity to identify diabetes (80% sensitivity) and a moderate capacity to identify hypertension (62% sensitivity) when the clinical data also indicated that they were present. Our findings align with studies in other populations, which indicate that well-defined, severe diseases requiring continuous care are reliably reported regardless of the data source [12,27].

Instead, moderate positive agreement with medical records was found for other comorbidities identified using the administrative definitions: autoimmune disease (43.5%) and chronic lung disease (31.3%). Nevertheless, the specificities remained consistently high, surpassing 90% (Table 2). Therefore, although these definitions may not reflect the prevalence of these comorbidities in the MS population due to limited sensitivity, they are unlikely to overestimate disease burden and can serve as valuable tools for disease surveillance.

With regard to the question of the best agreement indicator to use, in our study, PABAK, theoretically adjusted for prevalence, was in many cases found to be high, with a risk of producing misleading conclusions. Like other authors [28], we believe that no single agreement statistic can capture the desired information, but rather that different indicators (κ, positive and negative agreement, sensitivity, specificity, and positive/negative predictive value), offering different perspectives, can enrich agreement analysis.

## 5. Conclusions

Our results showed very good agreement between the administrative data and the clinical interviews for the detection of two conditions: diabetes and hypertension. For the other conditions analyzed, the administrative data showed modest sensitivity, particularly underestimating comorbidities such as mental health disorders or autoimmune diseases. Stratified analyses also revealed good agreement between data sources for chronic lung disease among male participants. Future research should aim to enhance comorbidity detection in individuals with MS by developing MS-specific algorithms that integrate multiple data sources, including outpatient and inpatient diagnostic codes, prescription records, and laboratory data.

These findings have potential implications for future health services research. As shown here and in prior studies, under-reporting of comorbid conditions in administrative databases is a considerable problem. The outcomes reported in these databases should therefore be interpreted cautiously, as they may have low sensitivity or could have under-reported or misclassified some comorbidities, particularly when health policy decisions are based on them.

## Figures and Tables

**Figure 1 healthcare-13-01281-f001:**
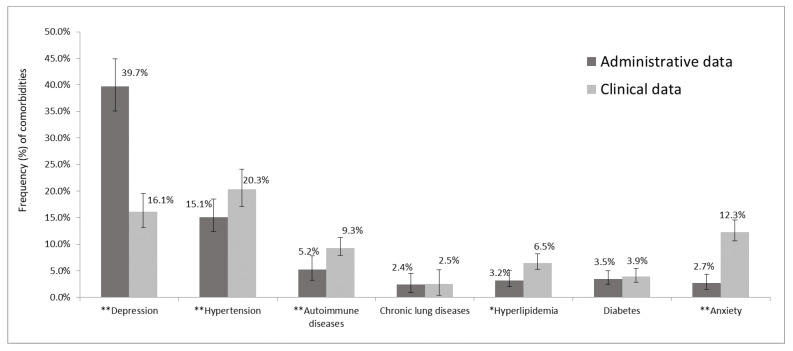
Frequency of major comorbidities in administrative and clinical data. *****
*p* < 0.05 and ** *p* < 0.001 by McNemar’s test; the bars represent the 95% CI of proportion comorbidities.

**Figure 2 healthcare-13-01281-f002:**
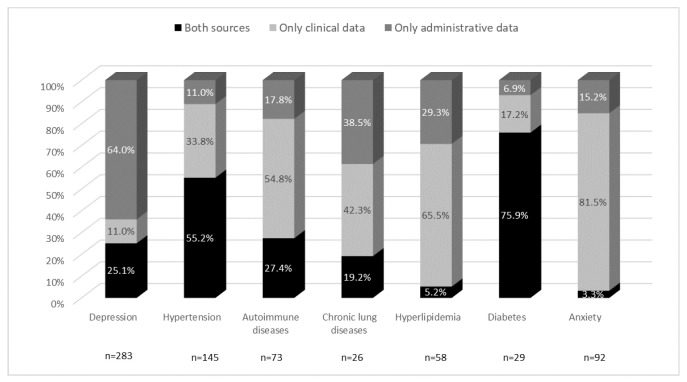
Major comorbidities captured by the different data sources.

**Table 1 healthcare-13-01281-t001:** Agreement between clinical and administrative data for major comorbidities (*n* = 635).

Comorbidities	Agreement	κ	BI	PI	PABAK	Positive Agreement	Negative Agreement
	(%)	(95% CI)			(95% CI)	(%)	(%)
Depression	66.60%	0.22 (0.15–0.29)	0.24	−0.44	0.33 (0.26–0.41)	40.10%	76.90%
Hypertension	89.80%	0.65 (0.57–0.73)	0.05	−0.65	0.80 (0.75–0.84)	70.80%	93.70%
Anxiety	85.90%	0.02 (−0.05–0.09)	0.1	−0.85	0.72 (0.66–0.77)	6.30%	92.30%
Autoimmune disease	91.80%	0.39 (0.26–0.53)	0.04	−0.86	0.84 (0.79–0.88)	43.50%	95.60%
Hyperlipidemia	91.30%	0.06 (−0.05–0.16)	0.03	−0.90	0.83 (0.78–0.87)	9.70%	95.40%
Chronic lung disease	96.70%	0.31 (0.09–0.52)	0	−0.95	0.93 (0.91–0.96)	31.30%	98.20%
Diabetes	98.90%	0.85 (0.73–0.96)	0	−0.93	0.98 (0.96–1.00)	83.30%	99.30%

κ = Cohen’s kappa; BI = bias index; PI = prevalence index; PABAK = prevalence-adjusted bias-adjusted kappa, CI = confidence interval.

**Table 2 healthcare-13-01281-t002:** Sensitivity, specificity, and positive (+) and negative (−) predictive values for administrative classification of major comorbidities compared with clinical interview (gold standard).

Comorbidities	Sensitivity	Specificity	Predictive Value (+)	Predictive Value (−)
Depression	69.60%	66.00%	28.20%	91.90%
Hypertension	62.00%	96.80%	83.30%	90.90%
Anxiety	3.90%	97.50%	17.70%	87.90%
Autoimmune disease	33.90%	97.70%	60.60%	93.50%
Hyperlipidemia	7.30%	97.10%	15.00%	93.80%
Chronic lung disease	31.30%	98.40%	33.30%	98.30%
Diabetes	80.00%	99.70%	90.90%	99.20%

## Data Availability

Data supporting the findings of this study are available upon request, subject to privacy restrictions due to the record linkage process based on tax codes. Upon request, the corresponding author will provide an anonymized Minimal Data Set (MDS) for sharing.

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
