# Peer review of "Capturing Information About Multiple Sclerosis Comorbidity Using Clinical Interviews and Administrative Records: Do the Data Sources Agree?"

_healthcare, 2025, doi:10.3390/healthcare13111281_

Round 1
Reviewer 1 Report
Comments and Suggestions for Authors
Thanks to the authors for presenting a very insightful study.
Here are my comments:
Overall, the paper is well written and organized with few to minor English language mistakes.
The topic is pertinent to the health informatics field – great research
- The introduction is concise but can be elaborated based on the newer techniques used - adding value of the current study e.g. line 66-70 need to elaborate on the previous research and evaluation outcomes with some more details.
- Materials and methods section is written very well, but the readers would want to see a complete detail of data integration and analysis. Line 86-89 mention variables but how the data are extracted, standardized vs non-standardized, etc. from the medical records under review. May add a few lines.
- The same can be applied to the “results” section which need to explain in terms of presentation of findings, demographic characteristics, and some table. I recommend if a life-table can be presented.
- In the discussion section lines 215 to 219 mention the clinical interviews. I would suggest incorporate data from the interviewees to make the model clear.
- Since line 226 and 227 mention the context of the clinical interviews, this would be an additional data source which the readers would look into.
- The discussion ends up with strengths and limitations provided systematically, but a sensitivity or subgroup analysis at the end of results would have supported the level of agreement and risk outcomes.
I would recommend if these points can be incorporated more or less, the manuscript is good for publication.
Best of luck to the authors.
Reviewer 2 Report
Comments and Suggestions for Authors
This study investigates the agreement between clinical interviews and administrative data in identifying comorbidities among people with multiple sclerosis (pwMS). The topic has strong relevance for both clinical care and public health, particularly in the context of secondary data use and real-world evidence. The study design is methodologically sound, employing multiple statistical indicators such as sensitivity, specificity, predictive values, Cohen’s kappa, and PABAK. Using clinical interviews as the reference standard adds credibility to the comparative approach.
Although the data were collected from only two Italian regions (Pavia and Liguria), the central question addressed—concordance between data sources—is broadly applicable across different healthcare systems. The findings are therefore of potential interest to a wide international readership, particularly those relying on administrative data for disease surveillance and population health management.
Several aspects of the study interpretation may warrant clarification or further elaboration, as detailed below.
- The authors employ clinical interviews supplemented by medical record review as the reference standard (“gold standard”) but do not explain how discrepancies between patient self-reports and medical documentation were handled during data extraction. Without a predefined adjudication process or hierarchy of information sources, this may introduce subjective bias. Further description of the procedures used to reconcile inconsistencies and ensure data reliability would strengthen the validity of the findings.
- In identifying mental health comorbidities, the algorithm used relies solely on prescription data based on ATC codes. However, antidepressants are often prescribed for off-label indications such as insomnia or chronic pain, which may result in substantial over-identification of depression or anxiety in administrative records. Depression in this study showed a sensitivity of 69.6% but a positive predictive value of only 28.2%, indicating a high proportion of false positives. The authors may consider potential improvements to the algorithm, such as incorporating diagnostic codes (e.g., ICD-9-CM), repeated prescription criteria, or psychiatric visit data to increase the precision of identification.
- The study uses ICD-9-CM as the coding system for administrative data during the 2021–2022 period. However, many healthcare systems have transitioned from ICD-9 to ICD-10-CM. Continued reliance on ICD-9-CM may limit the granularity and diagnostic precision, especially for complex conditions such as autoimmune or mental health disorders. The authors may acknowledge this as a limitation and discuss how the use of updated classification systems could improve data quality and specificity in future work.
- The analysis does not indicate whether key confounding factors were considered, such as patient age, disease duration, severity (e.g., EDSS score), or frequency of healthcare utilization. These variables are likely to influence whether comorbidity is documented in administrative data—for example, older or more severely affected patients may be more frequently coded due to increased healthcare utilization. The authors may discuss this limitation and, if feasible, consider stratified or multivariable approaches to account for potential bias.
- While the study highlights the limitations of administrative data in detecting certain comorbidities such as hypertension, autoimmune diseases, and anxiety, it does not offer specific recommendations on how to improve data integration or algorithmic design. It would be beneficial to propose actionable strategies, such as integrating outpatient and inpatient diagnostic codes, combining prescription and laboratory data, extending the observation window, or developing MS-specific comorbidity detection algorithms.
- The conclusion of the manuscript recommends interpreting administrative data with caution, which is valid but somewhat vague. Based on the stratified results, the authors could enhance the practical relevance of their findings by offering tiered recommendations—for instance, supporting the use of administrative data for diabetes and hypertension (where agreement is high), while recommending clinical data supplementation for depression or anxiety. Such distinctions would make the conclusions more actionable for health data users and policymakers.
- One of the key contributions of this study is its demonstration that administrative data vary in their suitability for identifying different types of comorbidities in pwMS. Specifically, the study shows that physiological conditions such as diabetes and hypertension are more consistently captured in administrative records, whereas psychiatric and autoimmune comorbidities are often underreported or misclassified. This differentiation between data types and conditions has practical implications for researchers and policymakers. To highlight this central contribution, the author may consider revising to strengthen the abstract—particularly the conclusion—to more clearly communicate the differential performance of administrative data across comorbidity types and to enhance the framing of the study’s implications for data-driven surveillance and research.
- One author (Simona Dalle Carbonare) is named in the author list and the conflict of interest statement but does not have a clearly stated contribution. It may be necessary to specify this author’s role in the Author Contributions section to ensure compliance with authorship criteria.
Reviewer 3 Report
Comments and Suggestions for Authors
This manuscript presents an important and relevant validation study; however, major revisions are needed:
Clarify methodology and validate tools (especially for mental health).
Deepen the critique of administrative data, particularly its weaknesses.
Refine statistical interpretation—do not overstate agreement based solely on PABAK.
Align conclusions with actual data, especially regarding utility for health policy.
The title is accurate but could be more specific. Suggest including the sample size or highlighting key findings (e.g., "limited agreement for certain conditions").
The abstract gives a fair summary, but omits limitations and methodological nuances. Include sensitivity limitations of administrative data and a clear statement on sample generalizability.
Line 61: Typo in "gahering" should be "gathering."
Lines 54–57: Strong justification for studying comorbidities, but lacks citation of any Italian-specific data to contextualize national relevance.
Lines 69–70: The sentence "there are only a few validated approaches…" is vague.
Suggestion: Cite examples or clarify what "validated" entails—statistical validation or clinical accuracy?
Lines 74–79: Important ethical details, but the study period (2020–2021) overlaps with COVID-19, which might affect comorbidity profiles.
Suggestion: Discuss potential COVID-era impact as a limitation or confounder.
Lines 81–83: Comorbidities were assessed for the "previous 10 years" — too wide a window without justification.
Suggestion: Justify time frame, especially for fluctuating or episodic conditions like depression/anxiety.
Lines 85–90: Inclusion of a CRF is excellent, but no clarity is given on how inter-rater reliability or training quality was ensured across interviewers.
Lines 106–110: The algorithm for depression/anxiety is ad hoc, raising validity concerns.
Suggestion: Validation references or internal validation metrics should be provided or discussed.
Lines 151–153: Over-reporting of depression in administrative data is stated but not explained until the discussion. Consider brief mention of possible reasons (e.g., antidepressants prescribed for off-label uses).
Lines 189–190: Cohen’s κ for anxiety is 0.02 — very poor. Authors attribute this to prevalence adjustment later but fail to directly criticize the data collection method.
Suggestion: Flag this in results as indicative of potentially flawed algorithms.
Table 1 & 2 (Lines 201–213):
Pos. predictive values for many comorbidities are very low (e.g., anxiety: 17.7%), which questions their use in policymaking.
Suggestion: Add a short results paragraph explicitly stating that for several conditions, administrative data lacks predictive utility.
Lines 223–224: The authors state that the interview approach is a hybrid between self-report and chart review. This distinction is confusing.
Suggestion: Clearly define if it involved direct verification or was still reliant on patient memory, which affects validity.
Lines 233–236: The limitation regarding generalizability is acknowledged, but the geographic limitation (two centers only) and sample demographics (64% female) could bias findings and should be discussed more.
Lines 246–251: Algorithm critique is valid, but insufficient.
Suggestion: Explicitly state whether this algorithm has been previously validated or piloted.
Lines 275–277: Authors mention that PABAK may be misleading but still use it to claim “excellent” agreement.
Suggestion: Consider removing overreliance on PABAK or at least tone down conclusions that use only this metric.
Lines 282–285: Claim that administrative records offer a "broad overview" seems overstated given how poorly they performed on five of seven comorbidities.
Suggestion: Rephrase to note that administrative records may help with surveillance of highly prevalent, well-coded conditions (e.g., diabetes) but are not adequate for mental health or nuanced diagnoses.
Line 289–291: Strong policy implications are made without sufficient caveats.
Suggestion: Add disclaimer about data quality variation across different health authorities and limitations of generalization to broader health systems.
Comments on the Quality of English Language
Proofreading is required
Round 2
Reviewer 2 Report
Comments and Suggestions for Authors
I would like to thank Dr. Ponzio and colleagues for their thorough and thoughtful revisions in response to my previous comments. The authors have fully addressed the concerns raised, including clarifications regarding data sources, potential methodological biases, and enhancements to the statistical analysis and interpretation of findings.
The study is methodologically rigorous, with a logical design and clear analytic approach. It effectively demonstrates the level of agreement between different data sources for various comorbidities in people with multiple sclerosis, and provides important insights into both the clinical and research implications of such findings.
The manuscript is well-structured and the discussion is insightful. Minor typographical and formatting issues (e.g., a repeated period, inconsistent spacing) are present but can be corrected at the proofreading stage and do not warrant further review.
Overall, I find the manuscript to be of high quality and recommend it for publication.
Reviewer 3 Report
Comments and Suggestions for Authors
The current version looks fine. I thank the authors for their effort in refining the manuscript. The manuscript may be acceptable for publication.